# Understanding the SARS-CoV-2 Virus Neutralizing Antibody Response: Lessons to Be Learned from HIV and Respiratory Syncytial Virus

**DOI:** 10.3390/v15020504

**Published:** 2023-02-11

**Authors:** Nigel J. Dimmock, Andrew J. Easton

**Affiliations:** School of Life Sciences, University of Warwick, Coventry CV4 7AL, UK

**Keywords:** human infection, repeated infection, SARS-CoV-2, HIV-1, RSV, neutralizing antibody, neutralization assay, epitope specificity, analysis of serum antibody specificity, monoclonal antibody

## Abstract

The SARS-CoV-2 pandemic commenced in 2019 and is still ongoing. Neither infection nor vaccination give long-lasting immunity and, here, in an attempt to understand why this might be, we have compared the neutralizing antibody responses to SARS-CoV-2 with those specific for human immunodeficiency virus type 1 (HIV-1) and respiratory syncytial virus (RSV). Currently, most of the antibodies specific for the SARS-CoV-2 S protein map to three broad antigenic sites, all at the distal end of the S trimer (receptor-binding site (RBD), sub-RBD and N-terminal domain), whereas the structurally similar HIV-1 and the RSV F envelope proteins have six antigenic sites. Thus, there may be several antigenic sites on the S trimer that have not yet been identified. The epitope mapping, quantitation and longevity of the SARS-CoV-2 S-protein-specific antibodies produced in response to infection and those elicited by vaccination are now being reported for specific groups of individuals, but much remains to be determined about these aspects of the host–virus interaction. Finally, there is a concern that the SARS-CoV-2 field may be reprising the HIV-1 experience, which, for many years, used a virus for neutralization studies that did not reflect the neutralizability of wild-type HIV-1. For example, the widely used VSV-SARS-CoV-2-S protein pseudotype has 10-fold more S trimers per virion and a different configuration of the trimers compared with the SARS-CoV-2 wild-type virus. Clarity in these areas would help in advancing understanding and aid countermeasures of the SARS-CoV-2 pandemic.

## 1. Introduction

Severe acute respiratory syndrome coronavirus 2 (SARS-CoV-2) is a newly arrived human pathogen, and there is much to be learned about it and how it interacts with its new host. This respiratory pathogen emerged in 2019 and most likely originated from a bat via an unknown non-human animal source. The virus causes serious mortality in people rendered susceptible by age, comorbidities, immunodeficiency or combinations of these problems, but it is less severe in healthy individuals. Vaccines expressing the main SARS-CoV-2 surface (S) envelope protein were rapidly devised and deployed in wealthy countries, and this included the first global use of an mRNA vaccine. These vaccines successfully reduced SARS-CoV-2-associated mortality, but, as with natural disease, antibody-associated immunity to subsequent infection was short-lived; the longevity of T cell-based immunity and its potential role(s) in the protection from subsequent infection are not fully understood yet. While vaccination importantly provides amelioration of disease symptoms, successive immunizations (up to five, to date) did not improve the longevity of protection, and even the combination of post-infection immunity and vaccination gave a similarly short-lived immunity to infection [1,2,3,4]. Thus, the human population as a whole is still at risk from an infection that ranges from the subclinical to a variable morbidity that has serious personal and economic impacts by requiring time away from the workplace. The impacts also include a debilitating condition known as ‘long COVID’, the causes of which remain unclear, although it has been linked with the reactivation of latent viruses such as Epstein–Barr virus [5]. An additional concern is that the cycle of repeat infections can transmit the virus to those in the high-risk group, particularly if the infection is inapparent or if people no longer feel it necessary to self-isolate. Added to these problems is the continuing evolution of new virus strains which afford some degree of escape from existing immune defenses, rendering an inadequate antibody response even more ineffective.

All the concerns about a short-lived, poor or inadequate immunity following infection or vaccination emphasize the need to understand the nature of the protective immune response against SARS-CoV-2 to better arm ourselves in the future against this and other viruses. This article considers the SARS-CoV-2-specific immunity that results from infection or vaccination. This appears to be largely antibody-based: sterilizing immunity cannot be mediated through T-cells. Space constraints mean that we focus on the direct neutralization of the SARS-CoV-2 by antibody. This should not be taken to mean that we place no value on other mechanisms in which neutralizing and non-neutralizing antibodies act against viruses via complement or other components of the immune system or that Fc receptors play a role in preventing or ameliorating disease; doubtless others will expand on these areas in due course.

The cause of the short-lived nature of the immunity is not known, although in the case of infection, it seems likely that SARS-CoV-2-virus-mediated immunosuppression/immunomodulation is involved. The scenario of initial protection followed by its decline suggests that cells responsible for producing antibody are not renewed, lose the ability to respond properly and be amplified or are ablated. SARS-CoV-2 has a large genome and expresses many proteins, and there is no shortage of candidate immunosuppressors. However, the mRNA vaccine expresses only the S protein, which would have to possess both immunostimulatory and immunosuppressive elements.

## 2. The Nature of Neutralizing Antibodies

While much is said about the importance of virus-neutralizing antibodies, an often-unstated key point is that they are not all equal in their ability to cause a loss of virus infectivity. A plethora of neutralizing antibodies that recognize different epitopes, and hence have different specificities, can potentially be elicited by any infectious agent or vaccine, and, in addition, the antibody response can vary according to the genetics of the host and their past infection experience. However, for reasons not understood, only a small fraction of the potential range of antibodies with these specificities may be expressed. The neutralization titer of (usually) a serum or plasma sample is not an absolute but represents the sum of the activity of all the neutralizing antibodies present in a particular sample. For example, this could consist of antibodies that are specific to many different regions of the target protein and hence will cause neutralization by interfering with the several different mechanisms that make up the infectious process—not only by blocking attachment to the main virus receptor binding domain (RBD), as is often assumed. Alternatively, the same neutralization titer could be achieved by a population that comprises one or only a few different antibody specificities [6,7,8]. Without analysis of the antibody specificities, it is not possible to distinguish between these two examples. With a simple assessment of a neutralization titer, they appear identical, but biologically, they are very different. In addition, antibodies to a particular neutralization region of the target protein can vary in the amount of each antibody specificity that is produced and in the efficiency with which each mediates neutralization. Thus, there is a huge permutation capable of qualitative and quantitative variability in the antibody population between individuals, and understanding the detail of this variability is key to understanding the factors that underpin an effective and long-lasting immunity.

## 3. The SARS-CoV-2 Surface (S) Protein

The SARS-CoV-2 S cell-attachment protein is anchored by a trans-membrane region at its C-terminus and undergoes proteolytic cleavage and structural rearrangements to form a metastable structure that is able to catalyze the fusion-entry process [9]. This is common to all enveloped viruses [10,11]. In the SARS-CoV-2 S protein, cleavage at the S1–S2 junction gives rise to an outer S1 subunit and an inner subunit (S2) that has a new N terminus. S1 and S2 are not covalently bound to each other. The binding of the S1 RBD to its receptor, the angiotensin-converting enzyme-2 (ACE2), in the cell membrane ACE2 triggers a second proteolytic cleavage of S2 at S2′ which exposes a new N terminus at the end of the 41-residue fusion peptide. Electron microscopy of SARS-CoV-2 virions shows that S1 exists in either a closed conformation (‘RBD down’) in which the RBDs are buried and unavailable for binding or an open conformation (‘1 or more RBDs up’) and appears to oscillate between the two positions [12,13]. The open conformation is required for the binding to ACE2 that leads to the fusion of the viral and cell membranes.

The number of fusion protein trimers on the surface of enveloped virus particles is very variable. SARS-CoV-2 (24+/−9 trimers of the S protein per virion) and HIV-1 (11–18 trimers of gp120-41 per virion) have a low number, whereas influenza virus A has an order of magnitude more (300–400 HA trimers per virion), all with similar-sized particles; respiratory syncytial virus (RSV) trimers are densely packed [14,15,16,17]. Curiously, influenza virus and SARS-CoV-2 trimers are not uniformly distributed and are not arranged in any discernible pattern [15,16]. Further, and perhaps uniquely, the SARS-CoV-2 trimers are hinged and can be tilted in any direction up to 50 degrees, rather than lying at right angles to the virion surface, as usually depicted. In addition, the number of S trimers present on a SARS-CoV-2 virion differs markedly from that of a VSV virion pseudotyped with a SARS-CoV-2 S protein which is commonly used to measure SARS-CoV-2-specific neutralizing antibodies, as VSV has 400 trimers per virion [18]. SARS-CoV-2 S protein-pseudotyped retroviruses, also used to measure SARS-CoV-2 neutralization, might more faithfully reproduce the SARS-CoV-2 S trimer distribution [12] and therefore be a more appropriate test system for evaluating SARS-CoV-2 neutralization.

A clear consequence of a small number of surface protein trimers per virion is fewer targets for neutralizing antibodies, as with HIV-1 [14]. How the distribution and/or number of trimers affect a SARS-CoV-2 neutralization event is not certain. It is not known for SARS-CoV-2 how many S trimers need to engage an antibody molecule to cause neutralization, and this is likely to vary with the function of the antigenic site in the infection process. If blocking the receptor binding domain (RBD) is the mechanism of neutralization, it seems unlikely that the binding of a single IgG molecule to one S trimer of the virion would be sufficient to cause neutralization, as this leaves up to 23 +/- 9 other trimers free to engage cell receptors. Indeed, rabies virus, closely related to VSV, could bind 130 molecules of a neutralizing monoclonal IgG (ca. 400 surface trimers per virion) without losing any infectivity [19]. The neutralization of a virion by a single antibody molecule would require interaction between all, or a sufficient number of, trimers and the triggering of some sort of cooperative event that renders the virion non-infectious—a situation for which there is, as of yet, no direct evidence. Additionally, it is not known whether a single SARS-CoV-2 S trimer can bind one, two or three molecules of IgG. The Mr of an S protein trimer ectodomain is approx. 540, and that of an IgG is approx. 160, so it might be possible to bind more than one IgG per trimer, depending on the position of the epitope and any spatial interference mediated by the mobility of the bound IgG. An S trimer construct is reported to simultaneously bind three Fabs of the monoclonal antibody (MAb) 4A8, but a Fab is approximately 30% of the mass of the whole antibody molecule and hence offers much less steric interference [20]. Unless trimers become clustered, they are spaced too far apart to be crosslinked by IgG, and the three repeated epitopes of a single trimer are too close together to allow for intramolecular crosslinking by an IgG molecule. However, the sparsity of S trimers on the virion surface and the flexibility of the S ectodomain allow for the maximum access of antibodies directed to any region of the S trimer. The sparsity of S trimers could also impinge on the efficiency of the fusion-entry process of SARS-CoV-2, as, for example, influenza virus requires the cooperation of three or more trimers for its fusion-entry process [21].

## 4. SARS-CoV-2-Specific Human Monoclonal Antibodies

Several groups have isolated SARS-CoV-2 S protein-specific monoclonal antibodies, e.g., [22,23]. Zost et al. isolated 389 SARS-CoV-2 S protein-specific monoclonal antibodies from two people who had been infected with the virus [22]. These were identified by a reaction with a stabilized prefusion form of the S protein ectodomain (1208 residues, with 6 residues substituted), an RBD construct (223 residues), or an S protein N-terminal domain construct (350 residues). Most MAbs, including those reacting with the N-terminal construct, were not neutralizing, but 67/70 neutralizing MAbs reacted with the RBD construct. In a further report, most of the 40 neutralizing MAbs tested blocked the interaction of the S trimer with the ACE-2 primary receptor [24]. Such data have led to the assumption that these MAbs neutralize by blocking the attachment of the virus to the host cell. However, blocking the interaction of individual proteins is a very different scenario from blocking a virion, which carries many copies of the target molecule and is many orders of magnitude larger, from attaching to the cell surface, and critical experiments clarifying the details of the neutralization process need to be conducted. One RBD-binding MAb (CR3022) has an entirely different mode of action and prematurely activates the post-fusion state of the S protein [25], although its epitope is distinct from the ACE-2 binding site [20]; a different MAb affects SARS-CoV-1 and MERS in the same way [26].

It would be strange if all SARS-CoV-2 MAbs were directed to the RBD region, as the S proteins of other enveloped viruses have several neutralization sites that are located on disparate regions of the protein. Other enveloped viruses, such as HIV-1 and influenza virus, with surface proteins that effect both attachment and fusion, stimulate antibodies which block the fusion-entry process [27,28]. For example, HIV-1 has six definably different antigenic sites with which neutralizing antibodies interact [29]. The apparent absence of SARS-CoV-2-specific neutralizing antibodies targetting regions other than the RBD is difficult to understand, as the selecting proteins, especially the prefusion form of the SARS-CoV-2 S ectodomain, contain the majority of the S protein trimer. However, there are several possible explanations: (a) it may be that the engineering required to stabilize the selecting protein for its use as an antigen alters its antigenicity. The conformational changes that result from cleavages that give rise to S1 and S2, and then S2′, are likely to be accompanied by the formation of new epitopes or an increased exposure of existing epitopes and, provided they are available for a sufficient time to activate B cells, to generate their cognate antibodies. However, antibodies to the fusion peptide of SARS-CoV-1 and SARS-CoV-2 exist in people who have recovered from infection and may be neutralizing [30,31], although assays confirming that fusion is inhibited were not reported. (b) Infection may bias the antibody response to the RBD region as seen with HIV-1 [32]. (c) MAbs were obtained from blood that was withdrawn at 50 days after infection, as this gave a better antibody response than blood taken at 35 days: thus, the time at which the blood sample was taken may be relevant [22]. This has been seen elsewhere; for example, the antibody response in Ebola virus infections may take up to one year to fully develop. (d) There may indeed be no other antigenic regions: however, this seems unlikely, as other viruses, such as HIV-1, as noted above, have multiple antigenic sites involved in other aspects of the entry process.

A final, but crucial, factor when extrapolating to antibody protection in vivo is the nature of the neutralization assay. As already mentioned, there is a concern regarding the relevance of assays based on pseudoviruses, such as VSV, which carry an order of magnitude more S trimers per virion than those that are present on the genuine SARS-CoV-2 virus [19]. In addition, it would be more relevant to use a competitive in vitro assay that mimics the in vivo situation in allowing for the simultaneous interaction of the virus, antibody, and host cells, rather than the conventional test in which preincubated virus-antibody complexes are added to susceptible cells. This is underlined by the high affinity of the S protein trimer for its receptor (in the µM range, whereas that of influenza virus is 1000-fold lower), suggesting that SARS-CoV-2 can escape neutralization by rapidly binding to and entering into a host cell [15]. It would be interesting to compare neutralization titers by the two methods suggested.

## 5. Mapping S-Specific MAbs That Neutralize SARS-CoV-2

Several studies describe the identification of SARS-CoV-2-specific highly neutralizing MAbs (e.g., [20,22,24]). Individual laboratories have made efforts to map their MAbs, but the key residues needed for epitope binding have been determined only for a few, usually through structural studies [20]. Binding assays usually use very large structures such as trimers of the S ectodomain, the RBD, and the N-terminal domain, which comprise many different epitopes. There is an urgent need in the global SARS-CoV-2 community for a comprehensive, interlaboratory mapping collaboration. In essence, the following conclusions emerge:

1. Highly neutralizing MAbs (N_50_ < 150 µg/mL) representing the RBD region and the N-terminal domain region have been found, but the frequency of occurrence and the number of such antibodies in body fluids during/after infection are not known, nor do we know if their production is stimulated by vaccination.

2. Most, but not all, MAbs (e.g., 47D11: reviewed by [20]) identified so far are inhibitory in the S trimer-ACE2 protein-binding assay, although, as noted above, this does not necessarily imply that these antibodies neutralize by preventing the attachment of the SARS-CoV-2 virus particle to the cell. Testing with authentic SARS-CoV-2 virus, not pseudovirions, is required to determine the mechanism(s) of neutralization.

3. There is clear evidence that MAbs that inhibit in the S trimer-ACE2 binding assay, and therefore impact the RBD, are diverse and interact with a number of different epitopes.

4. MAb P2B-2F6 competes in the S trimer-ACE2 binding assay and is shown by electron microscopy to bind to the RBD region in both its open (up) and closed positions, whereas ACE-2 binds to the RBD only in its open position. MAb P2B-2F6 has a higher affinity for the RBD than the ACE-2 receptor.

5. Many MAbs bind the N-terminal construct, but few are neutralizing; an exception is the highly neutralizing 4A8 MAb.

6. There is one report of post-infection antibodies that are specific for a peptide in the fusion region (mentioned above) [31].

7. Most studies show that the same selecting S protein constructs that identify neutralizing MAbs also reveal many non-neutralizing MAbs. While some of the latter could be antiviral in vivo through antibody-dependent cell cytotoxicity or other mechanisms, there is a concern that the ubiquitous non-neutralizing MAbs could block the binding and action of neutralizing antibodies and permit infection.

## 6. Comparison with Other Virus Systems

Here, we compare the SARS-CoV-2-specific antibody response with those found in other human virus systems—notably, human immunodeficiency virus type 1 (HIV-1) and respiratory syncytial virus (RSV). The HIV-1 field has been concerned with neutralizing antibodies for many years, and it now arguably represents the best studied and understood system of all. At first glance, it may seem strange to compare SARS-CoV-2 with HIV-1, as they cause such different infections and diseases, but both are enveloped viruses, and their major surface proteins, like those of all enveloped viruses, have a very similar structure. These are type 1 transmembrane homotrimer proteins with an external N-terminus and a C-terminus inside the virion, and they are cleaved to give an outer domain bound covalently or non-covalently, depending on the virus, to the inner domain. They all undergo profound structural rearrangements on contact with their primary cell receptor as part of the fusion-entry process.

## 7. What Can the SARS-CoV-2 Field Learn from HIV-1?

Like SARS-CoV-2, the HIV-1 and RSV fields are noted for the diversity of virus variants that differ in the sequence of their main surface protein, and they are classified into a number of clades, with multiple strains in each. It is an immense problem to devise a vaccine that can stimulate an immune response broad enough to protect against all variations of the virus. Particularly for HIV-1, this needs to be a sterilizing immunity, since, once infection is initiated, the HIV-1 genome is integrated into the DNA of the cell and cannot be removed. T-cell immunity is of little value, as it acts only on infected cells, so reliance has to be placed on the neutralizing antibody response. The HIV-1 field has trialled an unprecedented variety of neutralization immunogens and experimental hosts, mostly with modest results. Antibodies are made, but, as with SARS-CoV-2, these mostly exert an ineffective neutralizing response that comprises narrowly specific antibodies that can be directed against highly mutable epitopes; these antibodies may not prevent infection and are relatively short-lived. 

Another issue for HIV-1 was the assay system which, for convenience, used virus strains that had been adapted to grow in the laboratory—this adaptation requires the virus to mutate but, at the same time, renders it far easier to neutralize than a wild-type virus [33]. Thus, the laboratory-adapted strains are a poor indicator of protection in vivo. It is not clear if the SARS-CoV-2 system has the same problem, but it is suggested that adaptation to Vero-hACE2-TMPRSS2 cells may modify the S protein so that antibodies no longer neutralize as efficiently [23]. Second, there is concern over pseudotype assays that, for the convenience of scale and automation, are used as surrogates for the neutralization of SARS-CoV-2, as the distributions of S trimers on VSV and authentic SARS-CoV-2 are very different, as mentioned above [14,15]. The S protein-VSV pseudotype was variously shown to give neutralization that was statistically close to the neutralization of an infectious SARS-CoV-2 laboratory-adapted virus of that time [34] or was 21-fold more sensitive [22].

Eventually, the HIV-1 studies led to the discovery of broadly neutralizing monoclonal antibodies, which, as their name indicates, neutralize a plethora of different wild-type strains (31–99% of known strains) in vitro to a high titer (i.e., requiring from 3.7 to 0.002 µg/mL) [29]. Crucially, they also protect in vivo. The authors cite the macaque model, in which animals are given a natural mucosal challenge and develop an HIV-like illness, to show that 100% sterilizing protection could be achieved using a mixture of broadly neutralizing monoclonal antibodies and that this is related to the neutralization titer. Furthermore, they calculate that, for the HIV-1 infection of humans, a combination of four broadly neutralizing monoclonal antibodies at a serum concentration of 30 µg/mL could provide sterilizing immunity.

In summary, (a) experience with HIV-1 warns the SARS-CoV-2 field to question the validity of the pseudotyped virus used routinely in SARS-CoV-2 neutralization tests and to ask if the tests faithfully represent wild-type circulating virus; (b) like the SARS-CoV-2 vaccines, HIV-1 vaccines, in general, stimulate sub-optimal neutralizing antibody responses and fail to elicit the broadly neutralizing antibodies that are really needed for effective protection from infection; and (c) it needs to be determined if the human antibody repertoire is capable of responding to SARS-CoV-2 to produce the sort of broadly neutralizing antibodies that are so effective against HIV-1, and, if it is, a vaccine that can elicit them needs to be devised. Such antibodies have recently been found using a bioinformatic approach and include one that neutralizes the S2′ cleavage site [35].

## 8. What Can the SARS-CoV-2 Field Learn from the Respiratory Syncytial Virus?

Since its first description in 1956, RSV has been studied extensively to try to lessen the burden of the respiratory disease that it generates in seasonal epidemics, especially in the very young and the elderly. Several features of RSV infection and disease are similar to those found in SARS-CoV-2 infections, and the approaches to tackling these for RSV may provide useful insights into studies of SARS-CoV-2 with a view to developing therapeutic interventions. As with SARS-CoV-2, the robust antibody and cellular immune response to RSV infection that is detected following infection is short-lived. This, coupled with the appearance of genetic variants, means that repeat infections occur throughout life [36]. While the underlying processes leading to the short-lived immunity following natural infection are not entirely understood, it is known that RSV directly interferes with several aspects of the innate immune system, including the inflammatory and type 1 interferon responses as well as the intracellular processes involved in antigen presentation. These effects have consequences for the adaptive immune response which receives signals from the innate immune system, and the result is a reduction in or loss of B cell memory and T cell function, as reviewed in [37].

RSV has two envelope proteins: an attachment protein (G) and a fusion protein (F) that is present as a trimer in the prefusion state on the surface of infected cells and the virus particle; virus particles expressing only F are infectious, indicating that F is able to attach to host cells [38]. Initial studies of the neutralizing antibodies generated following infection with RSV were directed against the fusion (F) protein. However, neutralizing antibodies directed against both the pre-and post-fusion forms of the F protein are also present in human serum, and these interact with epitopes that are retained after the conformational change from pre- to post-fusion states [39,40]. No vaccine is currently available for RSV, though several are in clinical trials and have shown promising early results [41,42]. The only currently used treatment for RSV infection is a series of monthly injections of palivizumab, which is approved for prophylactic treatment in high-risk infants. Palivizumab is an F protein-specific, humanized mouse monoclonal antibody that prevents virus entry into cells. In recent years, modified versions of palivizumab have been generated to extend its half-life in the body following administration in order to reduce the need for monthly injections [43]. While palivizumab provides protection in approximately 50% of previously uninfected infants, it is not used for the treatment of infections in adults, where it is less effective. In recent years, it has become clear that it is necessary to better understand the immune response to infection to try to generate new approaches for prophylaxis and treatment [44].

Recent approaches have focused on investigation of the nature of the antibodies produced following natural infection. A study of 364 RSV F protein-specific monoclonal antibodies derived from memory B cells taken from three adult volunteers showed that these recognized a total of six different antigenic sites, including one not previously described [45]. The neutralizing anti-F protein antibodies with the greatest activity recognized the pre-fusion form of the F protein, and several showed 100-fold higher neutralizing activity than palivizumab, emphasizing the value of such studies for identifying potential prophylactic and therapeutic candidates. A similar study of 23 monoclonal antibodies isolated from human memory B cells confirmed this observation [46].

In a separate study, Andreano et al. investigated the repertoire of antibodies produced by single-cell sorted memory B cells from four adult volunteers. The data show that, while three subjects had generated antibodies specific for the pre-fusion form of the F protein, the majority of neutralizing antibodies were directed against the F protein bound to both the pre- and post-fusion forms. However, 78 of the 82 antibodies with dual-binding capacity were bound significantly more strongly to the pre-fusion form, which is found most predominantly on virus particles [47]. The data from these studies emphasize that the neutralization of virus infectivity in vivo is a potentially complex combination of a spectrum of different antibodies with different neutralizing activities. In the case of anti-RSV F protein antibodies, these are all targeted to epitopes closely located to each other on the F protein, so the relative concentrations of each, as well as their inherent affinity for their target epitope, will determine the competitive outcome of the interaction. In addition, these studies showed that many of the monoclonal antibodies have neutralizing activity against different genetic subtypes of RSV and, indeed, to the more distantly related human metapneumovirus, suggesting that they may exert broad range protection. The data also suggested that the genetic lineages of neutralizing antibodies seen in young children may differ from the lineages seen in adults. Such a bias in the immune response would be an important consideration for the design of vaccines for different target populations. It will be important to establish whether a similar situation exists with SARS-CoV-2 infections.

## 9. How to Better Understand the SARS-CoV-2 Neutralizing Antibody Response

To improve approaches for developing a strong, potentially long-lasting protective antibody response against SARS-CoV2, we need a measure of the individual antibody specificities that make up the neutralizing response produced to infection and vaccination—qualitatively in terms of their epitope specificity and quantitatively to determine their SARS-CoV-2-specific individual neutralizing titers, the breadth of neutralization, and the longevity of synthesis [48]. Crudely put, we need to know if the serum or plasma sample contains one or a few antibodies of high-titer specificities, antibodies of many different low-titer specificities, or a combination of both. In this way, we can analyze the response to a SARS-CoV-2 infection and determine the efficacy of a vaccine. This is needed to allow us to understand why the SARS-CoV-2 neutralizing antibody response is short-lived and ineffective in preventing infection. Only with this information can the SARS-CoV-2 neutralizing antibody response be properly evaluated, and through this, we will gain a better understanding of the deficiencies of the SARS-CoV-2 infection- and vaccine-induced neutralizing antibody responses and of why people can be infected repeatedly.

One way to analyze the SARS-CoV-2 neutralizing antibody response would be to assemble as extensive a library of SARS-CoV-2 S protein-specific monoclonal antibodies as possible, with emphasis on their specific neutralizing ability (µg required for 50% neutralization) and their capacity to neutralize a broad range of SARS-CoV-2 virus strains. Progress has already been made, but there is a worrying lack of MAbs that do not block the RBD. The main focus would then be to interrogate human blood/nasal wash antibody samples to see if these have the ability to block the binding of individual monoclonal antibodies to the authentic S protein. The blocking of binding would indicate the presence of a cognate MAb in the human sample or of one that has an overlapping footprint.

Alternatively, an epitope library for SARS-CoV-2 based on the sequence/structure of the S protein could be constructed. A linear peptide library is easily made, but peptides locked in a relevant conformation would likely be more useful. This could be approached using AI systems such as AlphaFold, which, in recent times, have revolutionized the determination of the 3D structures of proteins [49]. Each peptide would be synthesized with an anchor sequence to attach it to a substrate, and a library of such peptides would interrogate the antibody sample using an automated system. Basic quantitation could be provided by applying serial dilutions of the sample.

Electron- and cryo-electron microscopy are other methods that have been used successfully to map HIV-1 [50] and SARS-CoV-2 [20] surface protein-specific antibodies present in polyclonal sera after infection or vaccination. The traditional method of selecting neutralizing antibody escape mutants and determining which amino acid substitutions had occurred is available to those with biosecure facilities.

Other, but related, questions include why SARS-CoV-2 vaccines do not stimulate long-lived broadly neutralizing antibodies, and we need to know if such antibodies can be found in the human antibody repertoire. As for therapy, we need to know if such antibodies are to be found in other (possibly unconventional) animal systems and hence become a source for humanized monoclonal antibody reagents. Thinking outside the box led to the immunization of cows, which proved to be a surprising source of HIV-1-specific broadly neutralizing antibodies [51]. These have an ultralong CDRH3, which is a key feature of HIV-1-specific broadly neutralizing antibodies [29].

We have the space to quote only a limited number of papers; we apologize to the authors whose excellent work has been omitted.

## Data Availability

Not applicable.

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
