# Peer review of "Understanding the SARS-CoV-2 Virus Neutralizing Antibody Response: Lessons to Be Learned from HIV and Respiratory Syncytial Virus"

_viruses, 2023, doi:10.3390/v15020504_

Round 1

Reviewer 1 Report

This review/opinion article, authored by two experts in virus-specific antibody, highlights similarities and differences in the antibody responses to SARS-Cov-2 virus and those to HIV-1 and RSV in an attempt to shed light on the short-lived nature of the antibody response after SARS-Cov-2 infection or vaccination. They argue the need for understanding the performance of antibody at the individual epitope level and also raise concerns about how in vitro assays used to determine neutralisation capacity might not reflect in vivo activity.

The review is clearly written and relevant to the field. To my knowledge it is dissimilar to other reviews covering antibodies to this virus, thereby providing additional and important food for thought.  Although the review is  comprehensive, one improvement could be to provide more detail on what is known about antibody responses to SARS-Cov-2 in terms of their longevity.  How short-lived are these responses? What factors, such as severity of infection, have been found to influence this?  There are a small number of studies following antibody over time that are relevant to these questions and should be quoted (line 16-17 may have to be softened as a result).

Minor issues include rewording of the sentence line 95-96 eg. "Binding of the S1 RBD to its receptor, the angiotensin-converting enzyme-2 (ACE2), in the cell membrane triggers ..."

This is an important review that will be of considerable interest to researchers studying immunity to the virus.

Author Response

Reviewer 1

Many thanks for the very helpful suggestions. 

We have changed the sentence on lines 95-96 as suggested which has improved the clarity.

With regard to the issue of longevity of SARS-CoV-2 antibody response we take the point and have added some additional very recent information:

Lines 39-46: ‘These vaccines successfully reduced SARS-CoV-2-associated mortality but, as with natural disease, antibody-associated immunity to subsequent infection was short-lived; the longevity of T cell-based immunity and its potential role(s) in protection from subsequent infection is not yet fully understood.  While vaccination importantly provides amelioration of disease symptoms successive immunizations (up to 5 to date) did not improve the longevity of protection, and even the combination of post-infection immunity and vaccination gave a similarly short-lived immunity to infection [47-50].’

Reviewer 2 Report

Overall, this is a well written review focused on current limitations of developing neutralizing antibodies (nAbs) for prevention of SARS-CoV-2 with a comparison to HIV and RSV fields. However there are several limitations to the review that should be addressed.  The authors note the limitations of studies using pseudovirions (which differ in the number of trimers present on the viral envelope) but fail to address the importance of other antibody functions.  Specifically the importance of complement, which can modify the stoichiometry of nAbs and augment function is not addressed at all.  In addition the importance of FcR interactions to mediate antibody dependent killing is barely mentioned despite studies suggesting the importance of ADCC/ADCP.  The authors note that binding of Abs to viral envelope does not eliminate the virus (lines 124-126), which is precisely the argument why Abs that elicit antibody dependent killing either through ADCC/ADCP or complement dependent cytolysis is critical. This is mentioned briefly in lines 229-233 but inadequately developed.  There is strong data from both HIV and RSV field regarding need for FcR activity in addition to nAb responses.  

Additional specific comments:

Introduction:

General comment  The authors presume that sterilizing immunity should be the goal of mAb or vaccines. However, natural infections have been shown to boost vaccine responses and "sterilizing' immunity may not be the only or even optimal strategy.  Vaccines or Abs that prevent disease (rather than infection) might provide an advantage. This is exemplified by experiences with varicella-zoster.

Line 40-41.  While nAb titers do wane over time, the durability of T cell responses has not yet been fully defined and thus the notion of short lived immunity is more nuanced.  This sentence should be referenced and clarified.

Lines 44-45  The concept that long COVID is due to latent or persistent virus and the comparison to EBV is controversial. The incidence of long covid, which may be have complex causality, has declined with successive waves suggesting either that the immune response is protective and/or differences in viral variants and the host response to these variants. 

2, Nature of nAbs- lines65-85. This section needs to also include complement and its role in augmenting nAb responses. 

3. IN the discussion of RSV- should address the issue of antibody dependent enhancement. 

4. In the comparisons between SARS-CoV-2 and HIV, the limitations of the animal models for COVID should be included (relative to NHP for HIV).  This has also been a limitation for RSV (with reliance mostly on cotton rats).

.

Author Response

Reviewer 2

General comment: The authors presume that sterilizing immunity should be the goal of mAb or vaccines. However, natural infections have been shown to boost vaccine responses and "sterilizing' immunity may not be the only or even optimal strategy.  Vaccines or Abs that prevent disease (rather than infection) might provide an advantage. This is exemplified by experiences with varicella-zoster.

Reply: With respect we did not make any sweeping statement to the effect that sterilising immunity should be the goal of all vaccines.  Our comment was made specifically about HIV – for obvious reasons.  Non-sterilising immunity generated by the current SARS-CoV-2 vaccines has provided immense benefit in tempering the severity of disease.  We have amended the text to emphasise this.  However, non-sterilising vaccines leave the capacity for spread of the agent in the population.  This is a significant factor for respiratory viruses, although less so for agents where transmission is more easily controlled.  Thus, achieving complete protection from infection is an important goal for pathogens such as SARS-CoV-2.  Of course, such a goal may not be achievable but understanding the process is vitally important and that is one of the reasons we focus on neutralising antibodies in this review.

We have amended the statement that refers to the combination of post-infection immunity and vaccination giving a similarly short-lived immunity and together with a number of new references, have pointed out the value of current vaccines:

Lines 39-46: These vaccines successfully reduced SARS-CoV-2-associated mortality but, as with natural disease, antibody-associated immunity to subsequent infection was short-lived; the longevity of T cell-based immunity and its potential role(s) in protection from subsequent infection is not yet fully understood.  While vaccination importantly provides amelioration of disease symptoms successive immunizations (up to 5 to date) did not improve the longevity of protection, and even the combination of post-infection immunity and vaccination gave a similarly short-lived immunity to infection [47-50]. 

Specific comments

Lines 40-41 While nAb titers do wane over time, the durability of T cell responses has not yet been fully defined and thus the notion of short-lived immunity is more nuanced.  This sentence should be referenced and clarified.

Reply: We take the point about the durability of T cell responses and have added a comment to clarify as suggested by the reviewer: Lines 38-42 ‘These vaccines successfully reduced SARS-CoV-2-associated mortality but, as with natural disease, antibody-associated immunity to subsequent infection was short-lived; the longevity of T cell-based immunity and its potential role(s) in protection from subsequent infection is not yet fully understood.’

Lines 44-45 The concept that long COVID is due to latent or persistent virus and the comparison to EBV is controversial. The incidence of long covid, which may have complex causality, has declined with successive waves suggesting either that the immune response is protective and/or differences in viral variants and the host response to these variants. 

Reply: We accept this point and have tempered the comment accordingly:

Lines 46-52: Thus the human population as a whole is still at risk from an infection that ranges from the subclinical to a variable morbidity that has serious personal and economic impact by requiring time away from the workplace.  The impacts also include a debilitating condition known as ‘long Covid’, the causes of which remain unclear, although it has been linked with the reactivation of latent viruses like Epstein-Barr virus [1].

  1. Nature of nAbs- lines65-85. This section needs to also include complement and its role in augmenting nAb responses. 

Reply: We have focussed this review on the effects of direct interaction between neutralising antibodies and the virus rather than considering the additional mechanisms by which antibodies neutralise infectivity indirectly in conjunction with other agencies such as complement.  Space restraints were one consideration, but this omission should not be taken to mean that we think that direct antibody neutralisation is the only, or even most important process.  However, there is a paucity of information about these processes in the SARS-CoV-2 system.  We have added a statement to clarify the focus of the review while indicating to the readership that there remain other avenues for exploration:

Lines 61-64: ‘Space constraints mean that we focus on the direct neutralization of SARS-CoV-2 by antibody.  This should not be taken to mean that we place no value on other mechanisms in which neutralizing and non-neutralizing antibodies act against viruses via complement or other components of the immune system or Fc receptors play a role in preventing or ameliorating disease, and doubtless others will expand on these areas in due course.’

  1. In the discussion of RSV- should address the issue of antibody dependent enhancement. 

Reply: While ADE may be relevant, it comes out-with our focus for this review – as we have explained above: see new text lines 61-64.

  1. In the comparisons between SARS-CoV-2 and HIV, the limitations of the animal models for COVID should be included (relative to NHP for HIV).  This has also been a limitation for RSV (with reliance mostly on cotton rats).

Reply: This is indeed a major area, often controversial in terms of relevance to human disease.  Our review is entitled ‘Understanding the SARS-CoV-2 virus neutralizing antibody response’ which tacitly means the human neutralizing antibody response.  We have not discussed the merits and deficits of animal models for disease and related immunological studies as they are out-with the scope of this review. 

Round 2

Reviewer 2 Report

The authors have responded to prior suggestions.